

# An LSTM-based optimization algorithm for enhancing quantitative arbitrage trading

Guodong Han[1,2] and Hecheng Li[3]

[1] College of Computer, Qinghai Normal University, Xining, China
[2] Digital Finance Department, Bank of Qinghai, Xining, China
[3] School of Mathematics and Statistics, Qinghai Normal University, Xining, China

Corresponding author
Guodong Han,
hanguodong@stu.qhnu.edu.cn

## ABSTRACT

Arbitrage trading is a common quantitative trading strategy that leverages the long-term cointegration relationships between multiple related assets to conduct spread trading for profit. Specifically, when the cointegration relationship between two or more related series holds, it utilizes the stability and mean-reverting characteristics of their cointegration relationship for spread trading. However, in real quantitative trading, determining the cointegration relationship based on the Engle-Granger two-step method imposes stringent conditions for the cointegration to hold, which can easily be disrupted by price fluctuations or trend characteristics presented by the linear combination, leading to the failure of the arbitrage strategy and significant losses. To address this issue, this article proposes an optimized strategy based on long-short-term memory (LSTM), termed Dynamic-LSTM Arb (DLA), which can classify the trend movements of linear combinations between multiple assets. It assists the Engle-Granger two-step method in determining cointegration relationships when clear upward or downward non-stationary trend characteristics emerge, avoiding frequent strategy switches that lead to losses and the invalidation of arbitrage strategies due to obvious trend characteristics. Additionally, in mean-reversion arbitrage trading, to determine the optimal trading boundary, we have designed an optimized algorithm that dynamically updates the trading boundaries. Training results indicate that our proposed optimization model can successfully filter out unprofitable trades. Through trading tests on a backtesting platform, a theoretical return of 23% was achieved over a 10-day futures trading period at a 1-min level, significantly outperforming the benchmark strategy and the returns of the CSI 300 Index during the same period.

# INTRODUCTION

Quantitative trading, utilizing computer programs and mathematical models to execute trading strategies, represents an intelligent computational technique in the field of finance (*Chan, 2021*; *Liu et al., 2020*; *Yang et al., 2021*). Arbitrage strategy is a commonly used financial quantitative strategy, based on the fundamental principle that the price trends of multiple related financial assets are influenced by the same risk factors, thereby exhibiting a long-term equilibrium relationship (*e.g.*, stock price trends of companies like Intel and

AMD, or the price trends of futures contracts for iron ore and rebar steel) (*Dolado, Jenkinson & Sosvilla-Rivero, 1990*; *Enders & Siklos, 2001*). This long-term equilibrium relationship is known as a cointegration relationship and can be described by the stationarity of linear combinations of multiple price sequences. As the linear combinations of price sequences exhibit stationarity, they possess stable statistical properties characterized by mean-reversion features. Arbitrage strategies leverage this mean-reversion characteristic by executing a trade when the trend of their linear combination deviates from the equilibrium value, *i.e.*, buying undervalued assets and selling overvalued assets. When the trend of the linear combination reverts to the equilibrium value, the opposite action is taken, thereby generating profits.

With the advancement of quantitative trading technology, the prevalence of quantitative trading in various financial markets has increased significantly, making it increasingly challenging for trades based on the same strategy to be profitable in the market. On one hand, the cointegration relationship between price sequences of multiple assets is sensitive to price fluctuations, and even minor fluctuations in the market can disrupt the cointegration relationship, leading to frequent exits from trades and subsequent losses. On the other hand, traders employing the same arbitrage strategy often conduct numerous trades on the same assets, which can impact the cointegration relationship between multiple related assets, causing the trend of their linear combination to exhibit certain directional characteristics. In such cases, many strategies tend to trigger stop-loss actions, resulting in losses. This issue has become a challenging problem in quantitative trading (*Sezer, Gudelek & Ozbayoglu, 2020*).

Some studies have attempted to address this issue by incorporating trend identification strategies into traditional arbitrage trading strategies. They determine whether to temporarily exit trades when the cointegration relationship is disrupted based on whether a definite upward or downward trend appears in the linear combination of multiple assets (*Cao, Li & Li, 2019*; *Mbiti, 2021*). These studies employ conventional trend identification methods, such as the use of multiple moving average methods (*Hansun, 2013*), Bollinger Bands method (*Lauguico et al., 2019*), relative strength index method (*Panigrahi, Vachhani & Chaudhury, 2021*), among others. However, these conventional trend identification methods are not sufficiently accurate and are not well-suited for different types of assets.

Recent research has also explored alternative approaches to tackle this problem. For example, many researchers have proposed various methods based on transfer learning or deep learning to optimize basic arbitrage strategies (*Zhang et al., 2023*; *Nikoo, Khanagha & Mirzaei, 2023*; *Zhang, Zohren & Roberts, 2020*; *Chen et al., 2019*; *Jansen, 2020*). These optimization methods demonstrate greater robustness compared to traditional trend identification methods.

The article contributes to the field of quantitative trading by introducing a novel arbitrage strategy optimization model using long short-term memory (LSTM) networks. Key contributions include:

1. Developed a dynamic LSTM arbitrage (DLA) algorithm. Traditional arbitrage strategies, when using the Engle-Granger two-step method to determine cointegration

relationships, have strict conditions and are not sensitive enough to trends, leading to frequent stop-loss exits and losses, as well as difficulties in achieving profits through mean reversion when clear trends emerge. The new algorithm, by classifying price movements, assists the Engle-Granger two-step method in intervening with trading signals, overcoming the shortcomings of traditional methods and preventing arbitrage strategy failures.

2. The new algorithm optimizes trading boundaries based on the maximum value of trading frequency and single trade profit, effectively enhancing overall trading profits.

3. The effectiveness of the model has been demonstrated through backtesting, showing a significant improvement in theoretical returns compared to benchmark strategies and the CSI 300 Index.

The remaining sections of this article are organized as follows: "Related Works" discusses some works that are closely related to this study. "Problem Statement" describes the basic arbitrage strategy we designed, "The New Method" presents the new optimization algorithm we propose, "Experimental and Results Analysis" validates the effectiveness of our proposed algorithm by comparing it with benchmark strategy investment results through back testing historical data. Finally, "Conclusions" concludes the article.

## RELATED WORKS

In 2004, *Vidyamurthy (2004)* first proposed pair trading based on cointegration tests, laying the theoretical foundation for arbitrage trading. This method involves three key steps: selecting target assets, usually two correlated assets influenced by the same risk factors; using the optimized Engle-Granger two-step method for cointegration testing; and employing non-parametric methods to optimize entry and exit thresholds. Subsequent studies have proposed various optimization methods for arbitrage based on cointegration tests. *Lin, McCrae & Gulati (2006)* selected two Australian bank stocks for empirical analysis, adding a minimum profit constraint to the cointegration test method. *Caldeira & Moura (2013)* conducted a study using Brazilian stock market data, showing promising returns with low correlation to the stock market. *Clegg & Krauss (2018)* proposed the concept of "partial cointegration" and conducted empirical analysis, finding improved model characteristics.

Different from the cointegration method, another approach uses time series processing techniques to optimize arbitrage trading. *Elliott, Van Der Hoek * & Malcolm (2005)* built a basic theoretical framework for arbitrage trading using time series methods, employing a mean-reverting Markov chain to describe the spread process. Further improvements to Elliott's model were made, like *Do, Faff & Hamza (2006)* assuming log differences of stock prices follow an O-U process. *Chen, Chen & Chen (2014)* used regression GARCH time series principles for arbitrage trading models and conducted empirical analysis, surpassing previous models in terms of returns and feasibility. In 2019, *Huang & Martin (2019)* developed the ECM-DCC-GARCH model, showing better performance compared to the singular GARCH model.

In recent years, with the widespread application of machine learning methods, many scholars have also studied arbitrage trading optimization strategies based on neural networks and reinforcement learning.

*Sarmento & Horta (2020)* presents a novel approach to pairs trading using machine learning techniques. The study introduces a framework that combines principal component analysis (PCA) and the OPTICS clustering algorithm for efficient pair selection. Additionally, it explores a forecasting-based trading model using ARMA, LSTM, and LSTM Encoder-Decoder algorithms. The research demonstrates the effectiveness of these machine learning strategies in pairs trading, highlighting their potential to outperform traditional methods. The study utilizes a dataset of commodity-linked ETFs, emphasizing improved profitability and risk-adjusted returns.

*Kim & Kim (2019)* explores enhancing pairs-trading strategy using Deep Q-Network (DQN), a deep reinforcement learning method. This approach incorporates dynamic trading and stop-loss boundaries, optimizing the traditional pairs trading strategy which usually employs constant boundaries. The study tests the effectiveness of this optimized strategy against traditional methods, using stock pairs from the S&P 500 Index and demonstrating improved performance. The research suggests that deep reinforcement learning can significantly enhance financial trading strategies like pairs trading.

*Brim (2020)* explores the application of a deep reinforcement learning technique, specifically the Double Deep Q-Network (DDQN), in stock market pairs trading. The study focuses on using the DDQN to learn and predict the mean reversion patterns of cointegrated stock pairs, thereby facilitating profitable trading decisions. It introduces a negative rewards multiplier to adjust the system's risk-taking behavior during training. The research demonstrates the potential of reinforcement learning systems to effectively execute pairs trading strategies in the stock market and suggests future applications in other financial markets and trading strategies.

The effectiveness of methods based on traditional financial indicators is gradually diminishing due to their increasing use in quantitative trading, making it difficult to achieve sustained profits with trend judgment strategies that employ traditional financial indicators. Moreover, traditional methods treat the importance of long-term and short-term historical data equally, failing to flexibly handle the relationship between long-term and short-term data, such as increasing the weight of recent data in trend judgment. With the popularity of deep learning and reinforcement learning methods, LSTM networks have gradually achieved significant results in other fields due to their ability to automatically handle the relationship between long-term and short-term data. In the field of quantitative trading, existing research often directly uses trend prediction results as the standard for trading, which can easily fall into the trap of overfitting due to the complexity of market fluctuations. A better approach is to use LSTM networks as an auxiliary basis for trend judgment, while still obtaining trading signals using statistical arbitrage methods.

In summary, various methods exist to implement arbitrage trading strategies with differing effectiveness. While many optimization methods for arbitrage trading exist, few utilize LSTM models for trend classification and judgment, followed by optimization and filtering.

## PROBLEM STATEMENT

Here we explicitly state the assumptions and limitations of this study. The assumption involves two financial assets with strong correlations, including but not limited to stocks, futures, *etc.*, such as the stock price movements of Nvidia and AMD, or commodity futures of the same kind but with different delivery months, which are influenced by the same risk factors (such as supply and demand relationships), thus exhibiting strong correlations in their price movements. Statistically, a linear combination of the two prices exhibits characteristics of stationarity, known as the time series of the two prices satisfying a cointegration relationship (*Enders, 2015*; *Engle & Granger, 2015*). When a cointegration relationship exists, their price movements are closely aligned. If a significant deviation occurs, it may indicate that one asset is overvalued while the other is undervalued. At this point, arbitrage trading involves selling the overvalued asset and buying the undervalued asset to achieve a profit. This strategy is described in detail below.

Taking the price sequences of two assets influenced by the same risk factor as an example, let the price sequences of two strongly correlated financial assets be denoted as $x_t = (x_1, x_2, \ldots, x_n)$ and $y_t = (y_1, y_2, \ldots, y_n)$, then there must exist a linear combination of their prices:

$$z_t = x_t - \gamma y_t \tag{1}$$

that satisfies the stationarity condition. Here, $\gamma$ is the parameter of the linear combination. In this case, the linear combination has stationary lower-order moments (variance and mean) and exhibits mean-reverting behavior, meaning the variance and mean remain basically unchanged, as shown in Fig. 1.

In Fig. 2, the linear combination $z_t$ of the price time series $x_t$ and $y_t$ of the two assets demonstrates mean-reversion characteristics, causing the value of $z_t$ to fluctuate around its mean $\mu$. Based on the characteristics of the assets, the upper and lower volatility limits are set as represented by Formulas (2) and (3) respectively:

$$S_{upper} = \mu + \lambda\sigma \tag{2}$$
$$S_{lower} = \mu - \lambda\sigma \tag{3}$$

Here, $\lambda$ is asset-specific and can be set according to the volatility characteristics of its price time series. In this article, we employ a discrete dynamic approach for setting $\lambda$ values, which automatically obtains and updates the optimized $\lambda$ value based on the characteristics of the trading targets. The optimization method is detailed in "The New Method".

In Fig. 2, when the value of $z_t$ reaches or exceeds $S_{upper}$ (represented by the upper blue solid line in Fig. 2), it indicates that in the current market, the price of $x_t$ is overvalued, while the price of $y_t$ is undervalued. At this point, we sell asset $x$, and buy asset $y$. When $z_t$ reverts to the mean $\mu$ (represented by the blue dashed line in Fig. 2), we close the position to realize profits. Conversely, when the value of $z_t$ reaches or falls below $S_{lower}$ (represented by the lower blue solid line in Fig. 2), it implies that in the current market, the price of $x_t$ is undervalued, while the price of $y_t$ is overvalued. In this scenario, we buy asset x and sell

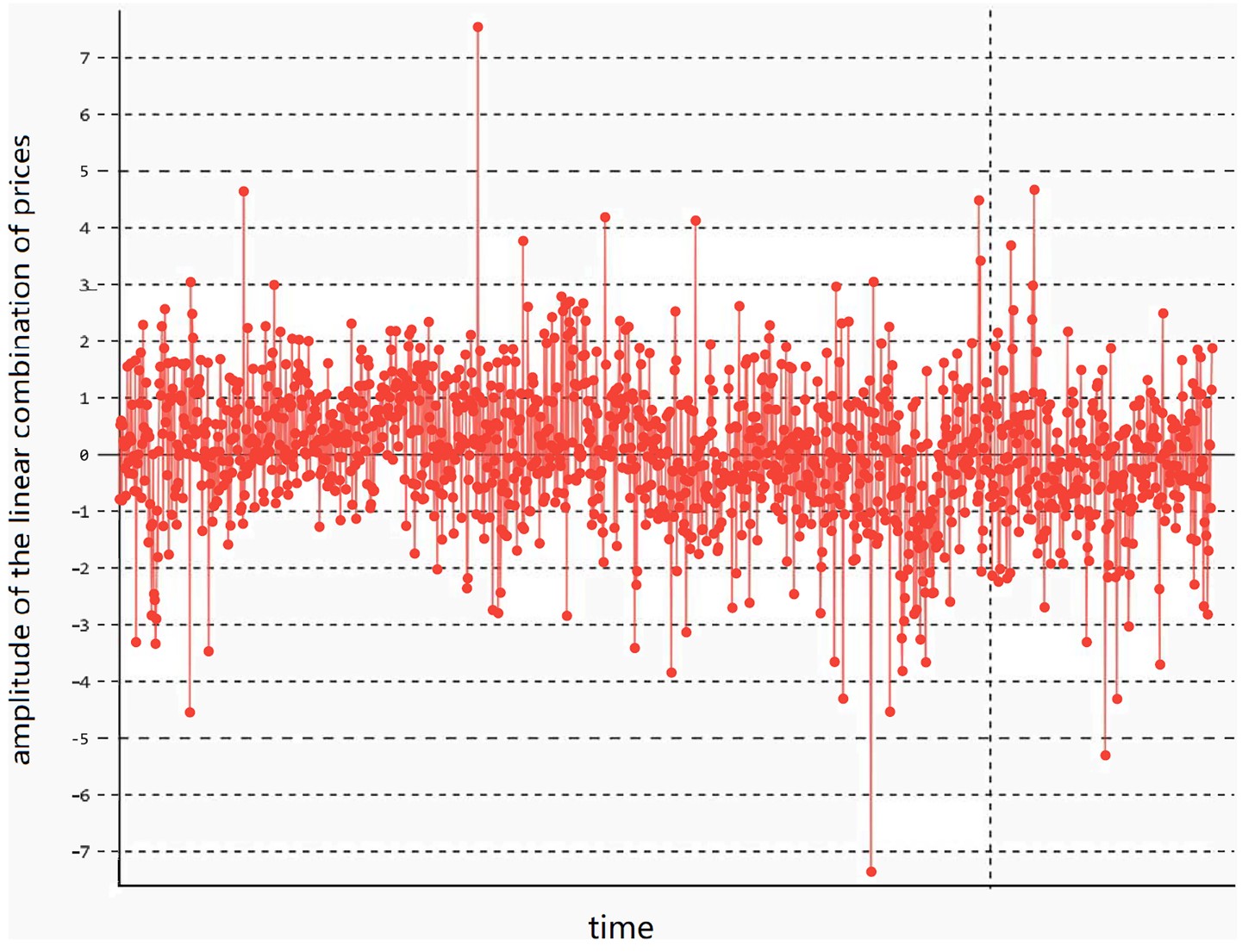

**Figure 1 The stationarity of the linear combination of the prices of two assets.** The trend of the linear combination after cointegration fitting of cross-period rebar futures RB1901 and RB1902 of the same variety.     

asset y. Upon $z_t$ reverting to the mean $\mu$, we close the position to realize profits. The trading process diagram for the aforementioned benchmark strategy is illustrated in Fig. 3.

To clearly define our problem, the definition of trend in this article is as follows (*Schwager, 1999*): An upward trend consists of a series of higher highs and a series of higher lows. That is, in the actual price series or moving average price series of a financial instrument, if a series of consecutive local maximum points gradually increase, and a series of consecutive local minimum points also gradually increase, then the price series is considered to be in an upward trend. Conversely, a downward trend consists of a series of lower highs and a series of lower lows. That is, in the actual price series or moving average price series of a financial instrument, if a series of consecutive local maximum points

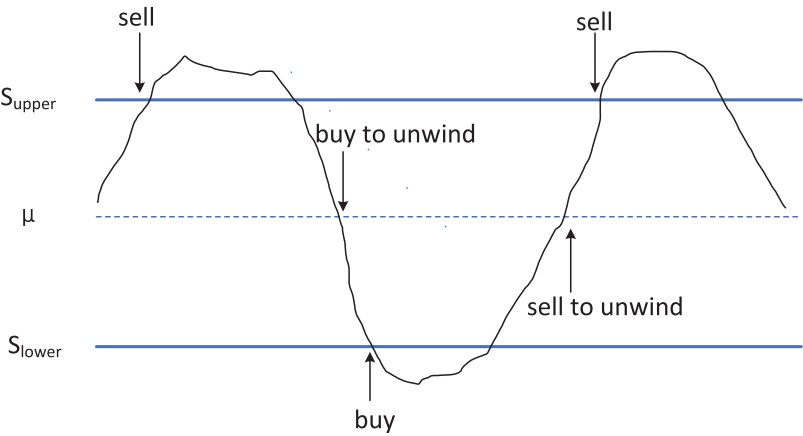

**Figure 2 Schematic diagram of utilizing cointegration relationships in arbitrage trading strategy.** The principle strategy of arbitrage trading based on cointegration relationships. When the price difference between two cointegrated assets reaches the upper limit (Supper, possibly referring to the upper threshold), traders will sell the higher-priced asset and buy the lower-priced one, anticipating that the future price difference will narrow back to long-term equilibrium. Conversely, when the price difference reaches the lower limit (Slower, possibly referring to the lower threshold), the trading strategy is to buy back the previously sold assets and sell the previously bought assets, thus profiting from changes in the price difference.

gradually decrease, and a series of consecutive local minimum points also gradually decrease, then the price series is in a downward trend. Movements outside of these trends are referred to as consolidation or fluctuation.

When the price sequences of two assets satisfy a strict cointegration relationship, meaning that the linear combination of their price sequences exhibits stationarity characteristics, the above-mentioned strategy can generate profits. However, when there is significant volatility in the futures market, for example due to the influence of supply and demand relationships in the spot market, one of the futures prices of the two assets may exhibit distinct volatility characteristics. In such cases, the cointegration relationship may be disrupted, and the linear combination of the prices of the two assets may exhibit clear upward or downward trends, no longer conforming to a Gaussian distribution. At this point, the above-mentioned strategy becomes ineffective.

To address this issue, we have trained a neural network model based on LSTM that can classify the trend of the linear combination of the two assets in advance. By doing so, it enables exiting the trade before the trend arrives, thereby avoiding losses.

## THE NEW METHOD

### Method overview

Based on the analysis in the previous section, the reason for the difficulty in sustaining profitability with the above strategy lies in the fact that the linear combination $z_t$ of price sequences of multiple assets exhibits certain trend characteristics, leading to slow or non-reverting mean reversion and triggering stop-loss actions, resulting in losses. There are multiple reasons for this phenomenon. For instance, the price of a certain futures asset may be influenced by short-term supply fluctuations in the spot market. When short-term

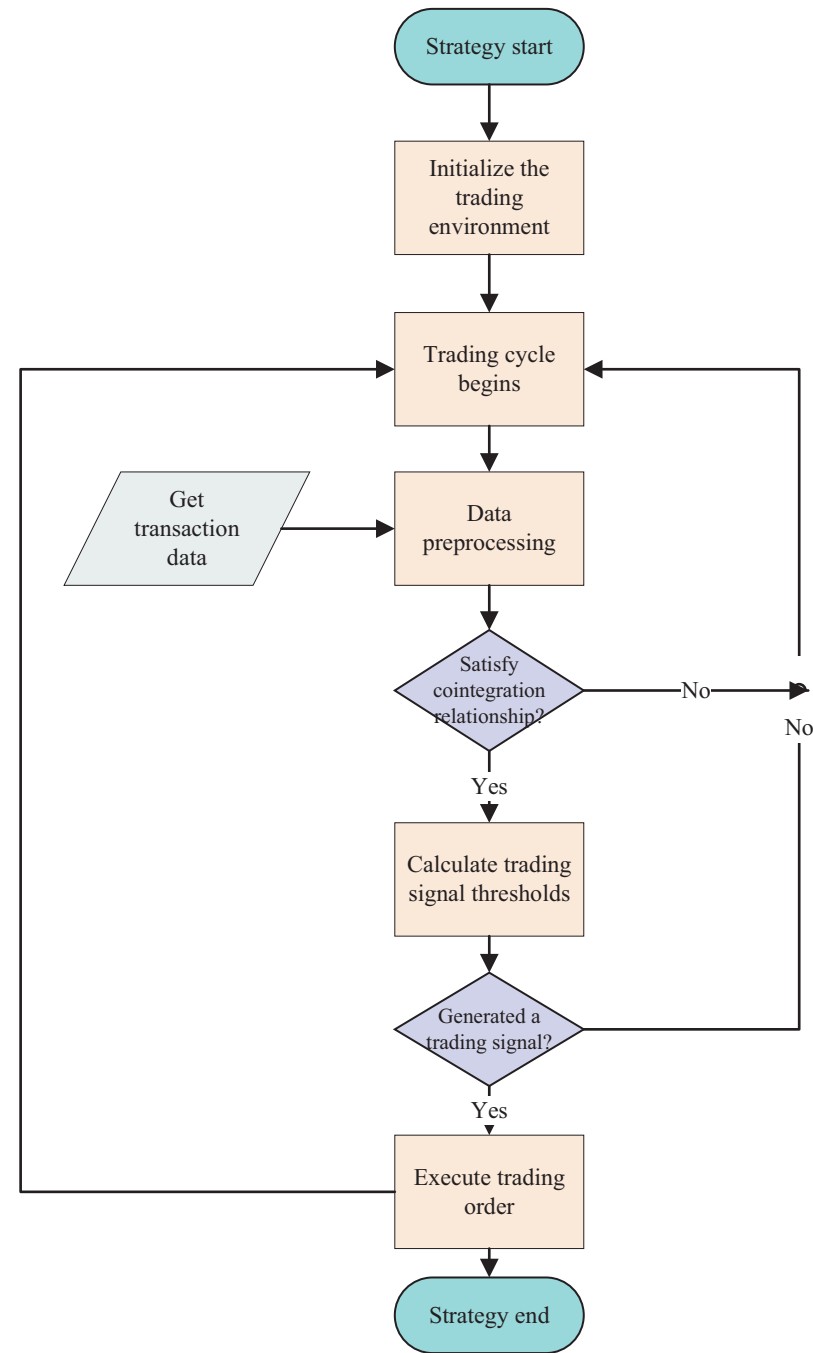

**Figure 3 Flowchart of the benchmark arbitrage strategy.** The arbitrage trading process based on the cointegration strategy, including data acquisition and preprocessing, determination of cointegration relationships, judgment of trading conditions, and execution of trades.

supply is restricted, it can cause significant increases in the futures price. Additionally, an excess of arbitrage trading in the market can also lead to significant fluctuations in futures prices. Quantitative strategies generally struggle to capture and handle market fluctuations

caused by macro factors, thereby leading to frequent losses in the benchmark arbitrage strategy.

To address this issue, we introduce the LSTM neural network method for trend classification of $z_t$. The LSTM model is a type of recurrent neural network model capable of handling time series data. Our proposed model can forecast the trend of $z_t$ in advance. If it is in a stable trend, arbitrage trading proceeds normally. When $z_t$ exhibits trend characteristics (clear upward or downward trends), we strategically exit the trade to filter out unfavorable trades and avoid larger losses. This forms the fundamental principle of the optimization algorithm proposed in this article.

The LSTM-based neural network model we constructed is capable of categorizing price movements based on historical data into upward trends, downward trends, and non-trend movements. The general workflow of the algorithm is as follows: First, we collect historical data for two futures and manually label segments of the data with upward and downward trends as well as consolidation (fluctuation) trends, dividing the labeled data into training and testing sets to train the neural network model for trend classification. In the strategy, the neural network model assists the cointegration judgment results to intervene in the trading signals. As previously mentioned, when determining cointegration relationships using the Engle-Granger two-step method, this method has strict conditions for the establishment of cointegration, often leading to the invalidation of arbitrage trading due to minor price fluctuations.

On the other hand, the cointegration relationship is not sensitive enough to trends, so if the Engle-Granger method concludes that the cointegration relationship still holds while prices are in an upward or downward trend, this can also lead to losses. Therefore, the LSTM-based neural network model can assist the Engle-Granger two-step method to intervene in trading signals. Specifically, when the Engle-Granger two-step method identifies a valid cointegration relationship, but the neural network model detects a trend, it is difficult to achieve mean reversion, making profitable arbitrage trading challenging and potentially leading to significant losses. In this case, trading is suspended and exited; when the Engle-Granger two-step method believes the cointegration relationship is disrupted, but the neural network model does not detect a significant trend, trading is maintained and awaits the appearance of a selling point.

The reason for this approach is that being overly sensitive to disruptions in cointegration relationships as per the Engle-Granger two-step method can lead to frequent exits from trades, causing losses. Meanwhile, in the presence of a trend, even if a short-term cointegration relationship exists, the absence of mean reversion characteristics can still result in losses.

On the other hand, existing arbitrage trading often uses fixed trading boundaries, SUPPER or SLOWER, as shown in Fig. 2. Their typical values are often a fixed multiple of the variance of the linear combination of the two assets' prices. This article proposes a dynamically optimized trading boundary algorithm that determines the trading boundaries dynamically based on the maximum value of the product of historical trading frequency and single trade profit.

**Table 1 Some key attributes of the trading data used.**

| Property name | Meaning |
| --- | --- |
| Open | Opening price |
| Close | Closing price |
| Volume1 | Asset 1 trading volume |
| Volume2 | Asset 2 trading volume |
| rsi | Relative strength index |
| macd | Moving average convergence divergence |
| hurst | Hurst exponent |
| ma10 | Moving average price with a period of 10 |
| ma20 | Moving average price with a period of 20 |
| ma30 | Moving average price with a period of 30 |

## Data collection and preprocessing

There are many ways to collect historical futures trading data, and many quantitative trading platforms also provide corresponding data access interfaces or download methods, which can be found in the descriptions of the quantitative trading platforms. This article uses the futures quantitative data interface provided by *JoinQuant (2024)*. In the JoinQuant quantitative trading platform, by providing the corresponding futures instrument codes and time range, one can obtain the relevant data for saving or direct use in the program.

In order to enable the LSTM model to accurately classify the trend characteristics of $z_t$, we have selected several trading attributes. Table 1 lists some of the attributes we have used.

Firstly, the data is manually calibrated. We identify prominent trend characteristics in the time series data of $z_t$ and label them accordingly. The endpoint of six consecutive price highs (where the next high is higher than the previous high) is established as the criterion for an upward trend, while the endpoint of six consecutive price lows (where the next low is lower than the previous low) is established as the criterion for a downward trend, with all other states considered as oscillating trends. The first 100 data points before the establishment of the trend are selected as training data. Data points indicating an upward trend are labeled as 1, those indicating a downward trend are labeled as −1. The remaining data is labeled as 0.

Due to the relatively short trading cycles and limited data volume for specific futures contracts, the quantity of data exhibiting trend characteristics in the linear combination of their prices is even scarcer. To address the issue of limited data, we augment the data by introducing Gaussian noise and salt-and-pepper noise. The method for introducing Gaussian noise to the closing price is illustrated in Formula (4).

Here, $x_{close}(t)$ denotes the closing price of asset $x$ at time $t$, $x'_{close}(t)$ represents the closing price after the Gaussian noise processing, α is the scaling coefficient, and $noise \sim N(0, \xi)$ denotes the noise.

$$x'_{close}(t) = x_{close}(t) + \alpha * noise \qquad (4)$$

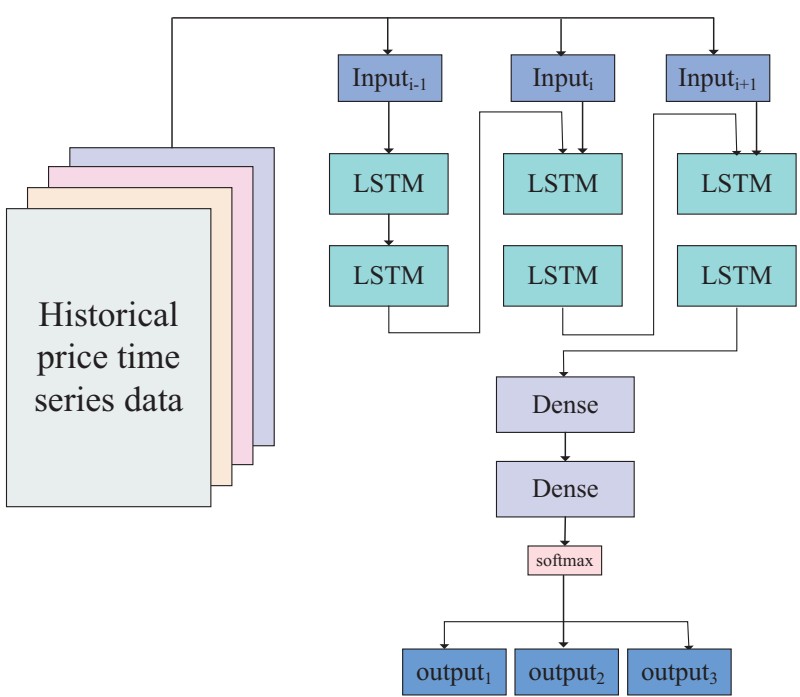

**Figure 4 Network model of dynamic-LSTM arbitrage.** The constructed neural network model, including two fully connected layers, two LSTM layers, and one softmax layer.

## Algorithm description

Based on the data size and empirical rules, we have designed a neural network model with a five-layer perceptron structure. The first two layers are LSTM layers, and the time series data of the two assets can be directly input into the first LSTM layer. After being processed by the two LSTM layers, the data enters two linear layers (fully connected layers), giving the data strong fitting capabilities. According to the calibration of the data in the previous section, the data outcomes consist of three categories (−1, 0, 1), representing upward trends, stable oscillation ranges, and downward trends, respectively. Therefore, the output of the neural network is ultimately classified into these three categories through activation by the softmax layer. The network model is illustrated in Fig. 4.

Using this LSTM-based neural network model, we classify the trends of the linear combination of $z_t$ values and filter out unfavorable trades based on the classification results. The optimization algorithm is presented in Algorithm 1.

Algorithm 1 requires historical and real-time trading data to drive its operation. Its input is a trained LSTM model, and its output is trading signals. In Algorithm 1, the first line of code initializes the trading environment, and the 2nd–28th lines are the trading loop.

In the trading loop, first obtain the historical and real-time trading price sequences of two assets and perform preprocessing and integration (lines 3–4). Preprocessing and integration mainly involve completing missing values and converting the original trading data into a format that the program can handle, such as filling in missing values. In line 5,

**Algorithm 1** Dynamic-LSTM arbitrage algorithm.

**Data:** Historical trading data and real-time trading data
**Input:** LSTM model
**Output:** Trading signals

1  Initialize the trading environment;
2  **while** *TRUE* **do**
3      Obtain historical_trading_data for two assets;
4      Preprocessing and integration of historical trading data;
5      $coint\_status \leftarrow coint\_test(historical\_trading\_data)$;
6      $trend\_status \leftarrow LSTM - Model(historical\_trading\_data)$;
7      Obtain real-time_trading_data for two assets;
8      Calculate the value of z as the linear combination of two assets;
9      **if** *position_empty* **then**
10          **if** *coin_status AND trend_status* **then**
11              **if** $z > z\_upper$ **then**
12                  Sell asset X and buy asset Y;
13              **else if** $z < z\_lower$ **then**
14                  Buy asset X and sell asset Y;
15              **else**
16                  continue;
17              **end**
18          **end**
19      **else if** *NOT coin_status AND NOT trend_status* **then**
20          Clear all positions;
21      **else**
22          **if** *X in longposition AND* $z < mean$ **then**
23              Clear all positions;
24          **else if** *Y in longposition AND* $z > mean$ **then**
25              Clear all positions;
26          **end**
27      **end**
28  **end**

carry out the unit root test method on the time series data of the two assets' prices to determine whether there is cointegration relationship. If the cointegration relationship is satisfied, the cointegration state is True; otherwise, it is False. In line 6, input the historical data into the LSTM model to determine the current trend status. If the current trend is ranging, the state value is 0; if the current trend is an upward trend, the state value is 1; if the current trend is a downward trend, the state value is −1.

In line 8, calculate the linear combination z value of the two assets' real-time prices. As mentioned earlier, the z value is a linear combination of the two prices. When the cointegration relationship is satisfied, the value of the time series should approximately satisfy the Gaussian distribution and be stationary. In line 9, determine whether the current position is empty. If it is empty, you can open positions. Lines 10–18 perform open positions operations, and judge whether the z value meets the opening conditions. When the z value is greater than the upper limit of volatility, sell asset X and buy asset Y. When the z value is less than the lower limit of volatility, buy asset X and sell asset Y.

Lines 19–20 are risk control operations. When the cointegration relationship no longer exists and the historical price sequence data of both assets has changed from ranging state to trend state, the trading condition has no longer been met, and it is necessary to close

**Algorithm 2** Transaction boundary optimization algorithm.

**Input:** Historcial trading data
**Output:** Optimized trading boundary
1  Let *lambdas* be an array from 0.5 to 4.0 with step 0.05;
2  Let *trading_times* be an empty array with the same length of *lambdas*;
3  **for** $i \leftarrow 1$ **to** *number_of_bars* **do**
4  $\quad$ Calculate the value of z as the linear combination of two assets;
5  $\quad$ **for** $j \leftarrow 1$ **to** *length_of_lambdas* **do**
6  $\quad\quad$ **if** $z > z\_upper$ $OR$ $z < z\_lower$ **then**
7  $\quad\quad\quad$ $trading\_times[j] \leftarrow trading\_times[j] + 1$
8  $\quad\quad$ **end**
9  $\quad$ **end**
10 **end**
11 $optimal\_lambda \leftarrow argmax(lambdas \times trade\_times - transactions\_fee)$;

positions and exit the market to avoid risks. Here, the "AND" connection of two parallel conditions is used because the sensitivity of the cointegration condition to small fluctuations in prices is relatively high; while the judgment of the trend also has a certain probability of being wrong, so by arranging the two conditions in parallel, it can avoid frequent liquidation and loss caused by frequent closing positions.

Lines 21–27 are stop profit and close position operations. When the linear combination z value of the two assets X and Y returns to the mean value, clear the position to get profits.

In Formulas (1) and (2), the value of $\lambda$ is used to determine the trading boundary ($S_{upper}$, $S_{lower}$). Its setting is crucial. When the value is set too large, the trading boundary is too wide, making it difficult to meet the trading conditions, leading to a lower trading frequency and a smaller order volume. When the value is set too small, the trading conditions are easily met, resulting in a narrower trading boundary and increased trading frequency. However, the profit margin for each trade is low, and may not even cover the transaction costs. Therefore, selecting an appropriate value for $\lambda$ is crucial.

Based on the optimization method for boundary values (*Zhao & Palomar, 2018*), for ease of implementation in the program, we design a discrete boundary value optimization algorithm to dynamically determine the appropriate value for $\lambda$. The principle of the algorithm is to determine the optimal $\lambda$ value from a series of discrete $\lambda$ values by maximizing the trading profit. The formula for calculating the optimized $\lambda$ is shown in Formula (5). The algorithm process is illustrated in Algorithm 2.

$$\lambda_{optimal} = argmax(\lambda_i * trade_{times} - transaction_{fee}). \tag{5}$$

Algorithm 2 can calculate the optimized trading boundary based on different assets. In Algorithm 2, the first two lines set up two arrays: the lambdas array represents all possible alternative $\lambda$ values from 0.5 to 4.0, which means the fluctuation range of the current asset linear combination is from mean ± 0.5 times the standard deviation to mean ± 4 times the standard deviation, as shown in Formulas (2) and (3). The trading_times array has the

same length as the lambdas array, and it is used to count the number of transactions that each candidate $\lambda$ value can trigger. When $\lambda$ takes a smaller value, the conditions for triggering transaction signals are relatively loose, so there are more transactions triggered but less profit per transaction; conversely, when $\lambda$ takes a larger value, the conditions for triggering transaction signals are relatively strict, and fewer transactions are triggered but more profit per transaction.

The for loop from line 3 to line 10 calculates the linear combination z value for each pair of values in the historical trading time series data of the two assets, and determines whether each candidate meets the trading condition based on the z value. When the trading condition is met, the corresponding $\lambda$ value's transaction count is increased by 1. This loop can count how many times each candidate $\lambda$ value triggers transactions in the current price time series data.

The 11th line is a broadcast operation. It calculates the optimal $\lambda$ value that can obtain the maximum profit by multiplying the transaction count of each $\lambda$ value by its single-transaction profit and subtracting the transaction cost required for trading. This optimized $\lambda$ value is related to the specific asset, and different assets have different optimal $\lambda$ values due to their different volatility characteristics. Moreover, the calculation of the optimal $\lambda$ value is dynamic and changes over time. Once the optimal $\lambda$ value is calculated, the optimal trading boundary can be calculated according to Formulas (2) and (3).

Through Algorithm 2, an optimized $\lambda$ value can be calculated using historical trading data when the trading strategy is initiated. This helps to achieve a balance between the number of transactions and single-transaction profits, thereby increasing trading profits.

## EXPERIMENTAL AND RESULTS ANALYSIS

*JoinQuant (2024)* is a quantitative investment research platform that provides the following functions: writing quantitative trading strategies based on Python, strategy backtesting based on historical trading data, simulated trading based on real trading data, *etc*. JoinQuant supports various financial instruments such as Chinese A-shares, exchange-traded funds, margin trading, commodity futures, and financial futures. It also provides a Python programming environment, data access APIs, strategy development tools, and more. The training of neural network models can be performed locally, with the local computer equipped with an i7-12700H processor, 32 GB of memory, and an Nvidia A2000 GPU. Backtesting experiments were completed on the JoinQuant platform. The specific software and hardware information of the local computer is shown in Table 2.

### Training of the LSTM neural network model

We used the RB1901 and RB1902 futures data from October 2018 to train the neural network, using 1-min period data. The neural network model training employed a learning rate of 0.0001, and the cross-entropy loss function was used due to the nature of the classification problem. During the training process, 100 sets of data were randomly

**Table 2 Local computer software and hardware information.**

| Name | Type/Version |
| --- | --- |
| CPU | 12th Gen Intel(R) Core(TM) i7-12700H, 2.30 GHz |
| RAM | 32 GB |
| GPU | Nvidia A2000 |
| Operation system | Microsoft windows 10 |
| Python | 3.11 |
| Numpy | 1.24.3 |
| Pandas | 2.0.3 |
| Pytorch | 2.1.0 |

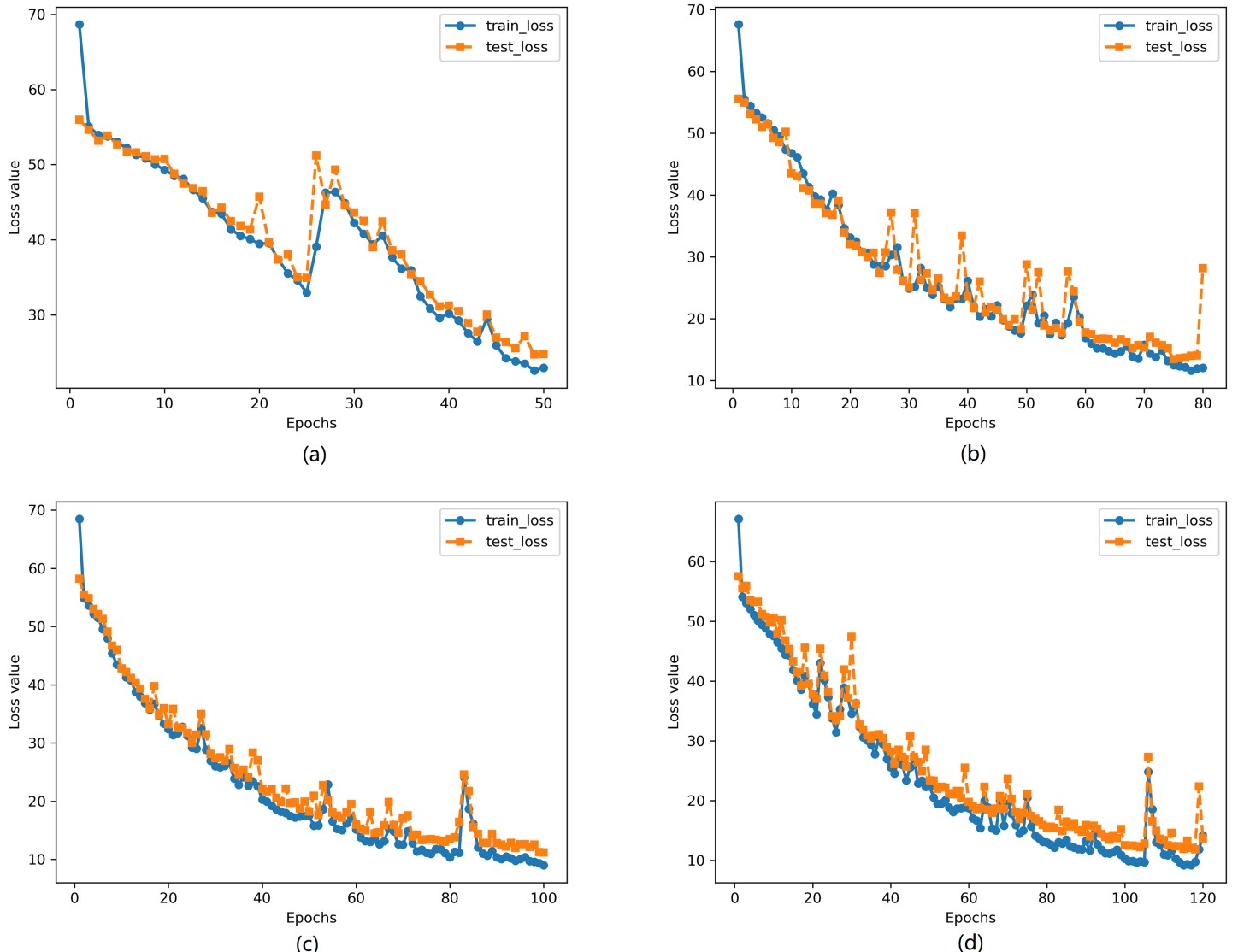

**Figure 5 Change of loss under different training epochs.** (A) Epochs = 50 (B) Epochs = 80 (C) Epochs = 100 (D) Epochs = 120.

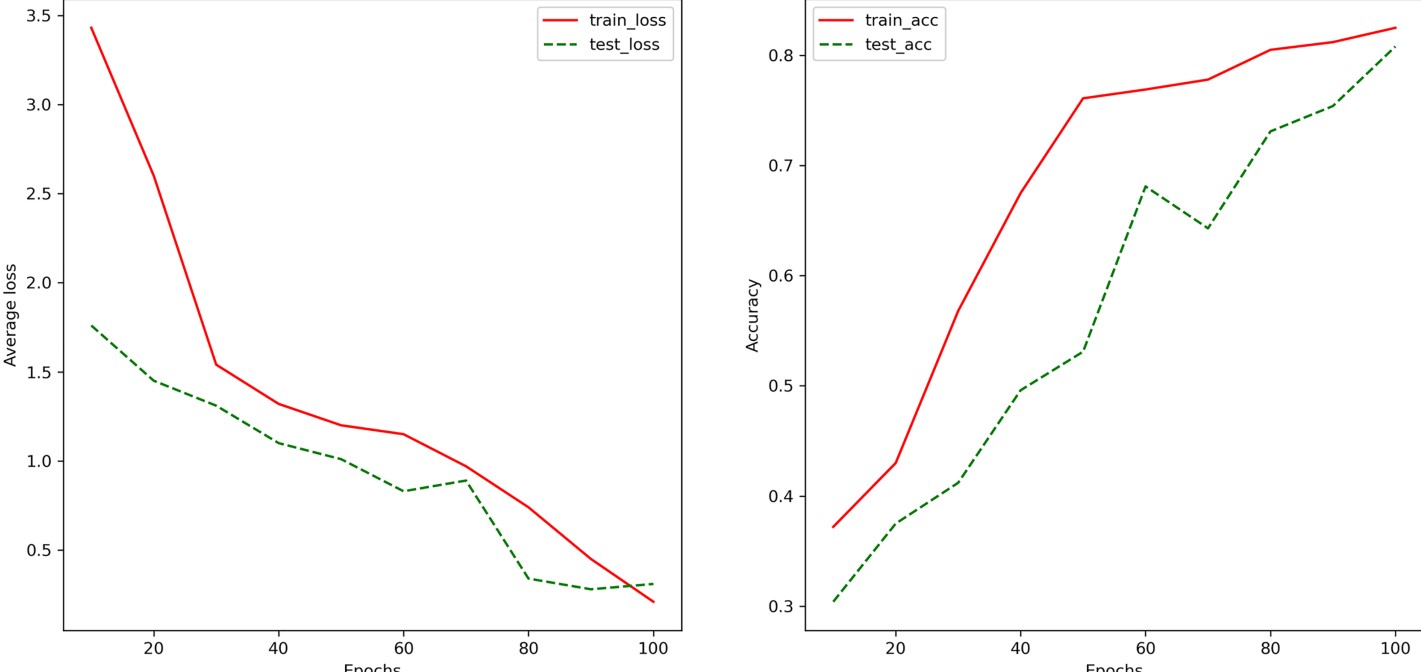

**Figure 6 The changes in the model's classification accuracy on the training and testing sets during the training process.** The changes in the accuracy of the LSTM-based neural network model on the training set and the test set as the training process progresses. It can be observed that the accuracy of the model gradually increases with the training process, indicating the absence of overfitting. The final model achieves an accuracy of 82.5% on the training set and 80.8% on the test set, implying that in arbitrage trading, when multiple assets exhibit significant trend characteristics in their linear combinations, the model is expected to halt trades with an accuracy of 80.8%, thus avoiding larger losses due to stop-loss strategies.

selected for batch training in each iteration. To observe the model's generalization ability and avoid underfitting and overfitting, we used four different training epochs: 50, 80, 100, and 120. The results of the neural network model training are shown in Fig. 5.

Figure 5 shows the changes in average loss of the model on the training set and test set during the training process. It can be observed from the figure that the loss of the model decreases significantly with the increase in training epochs.

Figure 6 depicts the changes in the accuracy of the LSTM-based neural network model on the training set and the test set as the training process progresses. It can be observed that the accuracy of the model gradually increases with the training process, indicating the absence of overfitting. The final model achieves an accuracy of 82.5% on the training set and 80.8% on the test set, implying that in arbitrage trading, when multiple assets exhibit significant trend characteristics in their linear combinations, the model is expected to halt trades with an accuracy of 80.8%, thus avoiding larger losses due to stop-loss strategies.

Figure 7 shows a partial classification result of the LSTM model. In a scenario of three categories, the figure uses red, yellow, and green to classify uptrend, consolidation trend, and downtrend respectively. The figure indicates that the three categories generally align with the observed results.

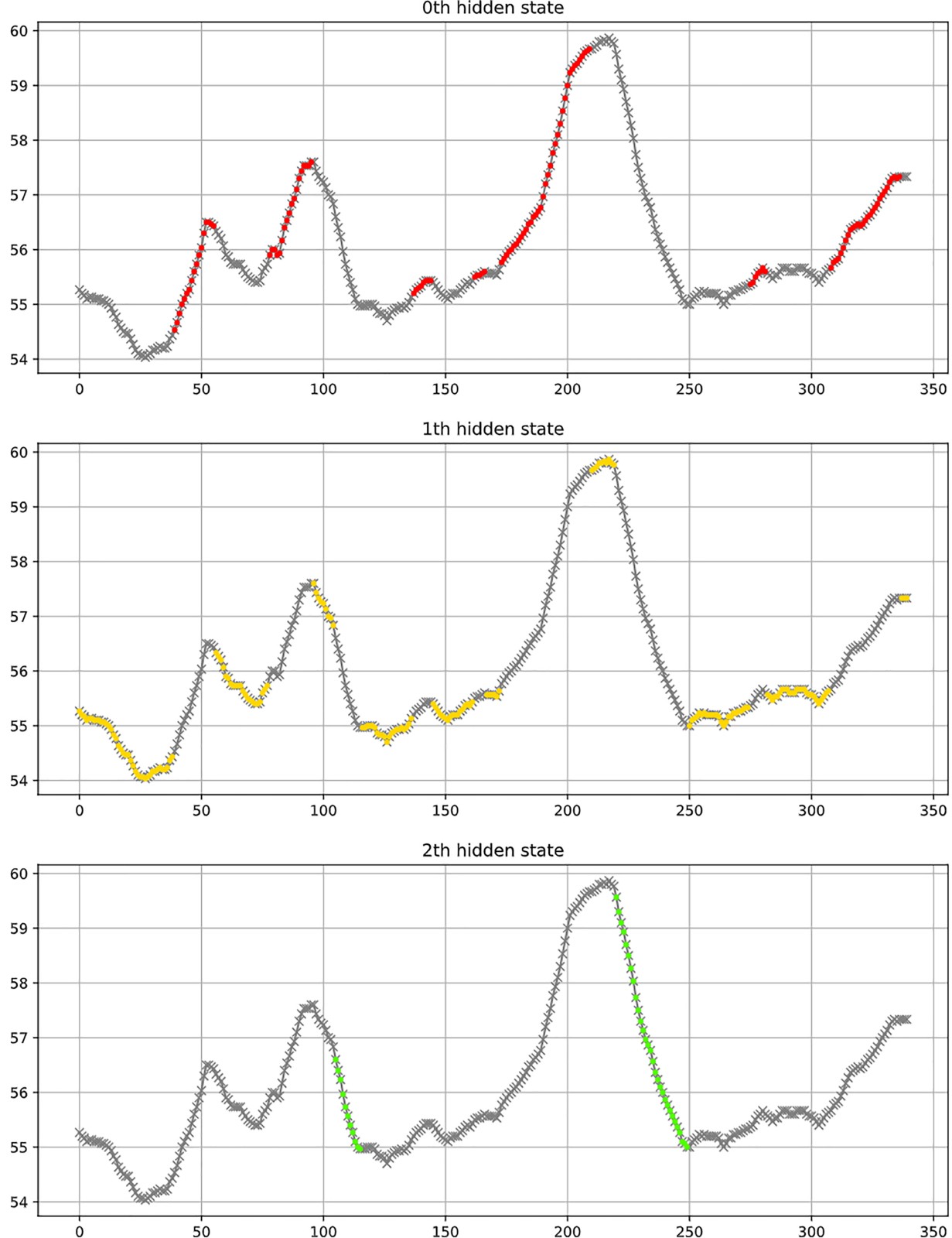

**Figure 7 Example of classification results, where red represents an uptrend, yellow indicates a consolidation trend, and green signifies a downtrend.**

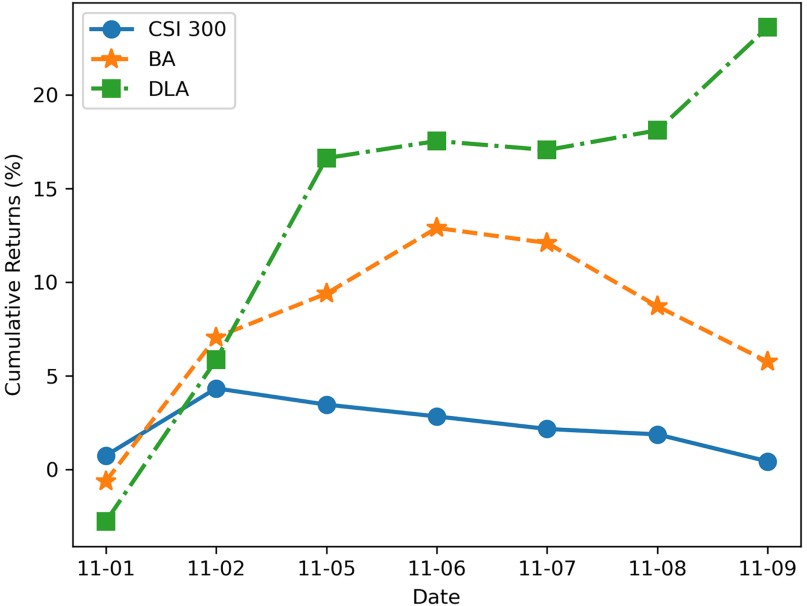

**Figure 8 Cumulative return rate backtested on JoinQuant platform.**

## Backtesting experimental results

We utilized the JoinQuant platform (*JoinQuant, 2024*) for testing the profitability of the new DLA algorithm. Additionally, we used the benchmark strategy (BA) described in "Problem Statement" and the profit curve of the CSI 300 index during the same period for comparison. Both strategies were backtested using the trading data of RB1901 and RB1902 from November 1, 2018, to November 10, 2018 (7 trading days in total), with a data frequency of 1 min and an initial trading amount of 1 million RMB. The test results are illustrated in Fig. 8. It can be observed from Fig. 8 that the strategy enhanced by the new algorithm exhibits a significant improvement in profitability, effectively avoiding losses in arbitrage trading caused by trends.

The results shown in Fig. 8 demonstrate the advantages of the LSTM-based algorithm. The reason for the algorithm's better performance is that when the benchmark algorithm uses the Engle-Granger two-step method to judge cointegration relationships, the conditions for establishing cointegration are quite strict, resulting in a higher frequency of cointegration not being established, which leads to frequent stop-losses and losses in trading. On the other hand, the new algorithm classifies trends, and when the price movement is in an upward or downward trend, it is difficult to achieve mean reversion at this time, so exiting the trade avoids greater losses. On the other hand, optimizing trading boundaries also contributes to enhancing overall trading profits.

It is important to note that the trading on the JoinQuant platform and actual trading may not be entirely consistent, as there can be differences in the execution of transactions and the matching algorithms compared to the real environment. Therefore, while the returns in Fig. 6 may differ from real market returns, they still reflect the difference in profitability between the new algorithm and the benchmark strategy.

## CONCLUSIONS

While the proposed Dynamic-LSTM Arb (DLA) strategy demonstrates promising results in enhancing quantitative arbitrage trading, this study acknowledges several limitations that may influence the generalizability and applicability of the findings.

Firstly, the effectiveness of the DLA strategy is tested under specific market conditions and with a selected dataset comprising of futures trading data for hot-rolled coil and rebar steel. Therefore, the performance of the DLA strategy in different market conditions, with other financial instruments, or across diverse time periods may vary and requires further investigation.

Secondly, the study assumes a strong correlation between the two financial assets under consideration, such as the stock prices of Nvidia and AMD, or commodity futures of the same kind but with different delivery months. This assumption may not hold in all trading scenarios, especially in volatile or rapidly changing markets where the correlation between assets can weaken or become unpredictable over time.

Furthermore, the implementation of the DLA strategy relies heavily on LSTM neural networks' ability to classify and predict trend characteristics accurately. While the model achieves an accuracy of over 80% in trend classification, there is an inherent limitation in using machine learning models, including the potential for overfitting to historical data and the challenge of capturing the full complexity of market dynamics.

Lastly, the study does not fully explore the impact of transaction costs, slippage, and market liquidity on the profitability of the arbitrage strategy. These factors can significantly affect the net returns of trading strategies in real-world scenarios and should be considered in future research to provide a more comprehensive assessment of the DLA strategy's viability.

### Funding

This work was supported by the National Natural Science Foundation of China under Grant No. 61966030, the Qinghai Province Key Laboratory of the Internet of Things Project (Grant No. 2022-ZJ-Y21), and the Qinghai Provincial High-End Innovative and Entrepreneurial Talents Project. The funders had no role in study design, data collection and analysis, decision to publish, or preparation of the manuscript.

### Grant Disclosures

The following grant information was disclosed by the authors:
National Natural Science Foundation of China: 61966030.
Qinghai Province Key Laboratory of the Internet of Things Project: 2022-ZJ-Y21.
Qinghai Provincial High-End Innovative and Entrepreneurial Talents Project.

### Competing Interests

Han Guandong is employed by Bank of Qinghai.

## Author Contributions

- Guodong Han performed the experiments, analyzed the data, performed the computation work, prepared figures and/or tables, and approved the final draft.
- Hecheng Li conceived and designed the experiments, authored or reviewed drafts of the article, and approved the final draft.

## Data Availability

The code is available in the Supplemental File.

## Supplemental Information

Supplemental information for this article can be found online at http://dx.doi.org/10.7717/peerj-cs.2164#supplemental-information.

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
