# Peer review of "An LSTM-based optimization algorithm for enhancing quantitative arbitrage trading"

_PeerJ Computer Science, doi:10.7717/peerj-cs.2164_

## Round 0.1 · original submission · Major Revisions

As per comments from three reviewers, I suggest a major revision for further consideration. In the revised paper, the authors should follow all comments from reviewers carefully, then it may be accepted.

**Language Note:** The review process has identified that the English language must be improved. PeerJ can provide language editing services - please contact us at [email protected] for pricing (be sure to provide your manuscript number and title). Alternatively, you should make your own arrangements to improve the language quality and provide details in your response letter. – PeerJ Staff

Reviewer 1 ·

Basic reporting

To address the uncertainty of arbitrage trading, this paper proposes an optimization strategy based on dynamic LSTM Arb. The overall content of this article is disorganized, with confused logic, and insufficient experiments. Therefore, this paper is not suitable for publication in this journal. There are some general suggestions to help you.
1. I suggest that the author add numbering to each heading.
2. In the second section, the author only states the methods of arbitrage trading without elaborating on their drawbacks. I suggest that the author discuss the limitations of these approaches in the second section.
3. The descriptions and captions of the figures in the article are somewhat disorganized. I suggest that the author reorganize the captions and descriptions of all the figures in the text.
4. For the convenience of readers, I suggest that the author place the figures and tables in their respective chapters.
5. There are some grammar and spelling errors in the article. I suggest that the author carefully check and revise them. For example, in line 172, '(z_t)' should be corrected.
6. The content of the three subsections in "The new method" section is confusing. I suggest that the author reorganize the content of this section.
7. I suggest that the author provide information about the experimental environment.
8. I suggest that the author provide the necessary information for the experiments, such as learning rate, loss function, number of epochs, and so on.
9. I suggest that the author provide an introduction to the dataset used in the article, including its quantity, source, and how it was used in the experiments.
10. The epochs in the four subfigures of Figure 5 are different. Is this a normal phenomenon? Please the authors explain this issue.
11. For the rigor of the experiment, I suggest that the author add comparative experiments between this method and other researchers' methods.

Experimental design

no comment

Validity of the findings

no comment

Additional comments

no comment

Cite this review as

Reviewer 2 ·

Basic reporting

In this manuscript, the authors propose an arbitrage strategy optimization model built on Long-Short-Term Memory networks. The new arbitrage strategy established based on this model and combined with the benchmark arbitrage strategy showed significantly higher profits in the back-testing trading of the historical data of hot-rolled coil and rebar steel over a 10-day period, proving the effectiveness of our proposed algorithm. The authors suggested addressing the following comments and suggestions when preparing the revised version:
= Abstract: The section needs to be re-drafted to be self-contained, which means it has to clearly show the hypothesis, methodology, techniques, and tools used, and the results obtained.
= Keywords: Authors suggested to update the keywords by selecting more relevant terms. Keywords play an important role in the appearance of the manuscript in scholars' searches, which will give it more hits and more citations.
= What assumptions did the authors make during the simulation phase of this research work? If there is any.
= What limitations did the authors face during this research work? If there is any.
= Authors suggested updating the introduction and the related work sections by including some of the most recent published works.
= Conclusion: The conclusion should be abstracted, so authors need to consider re-drafting it.
= Authors need to confirm that all acronyms are defined before being used for the first time.
= Authors need to confirm that all mathematical notations are defined when being used for the first time.
= The Authors suggested proofreading the manuscript after addressing all comments to avoid typos, grammatical, and lingual mistakes.

Experimental design

.

Validity of the findings

.

Additional comments

.

Cite this review as

Reviewer 3 ·

Basic reporting

The manuscript entitled “An LSTM-based optimization algorithm for enhancing quantitative arbitrage trading” has been investigated in detail. The manuscript suffers from vague terminology, lack of specificity, and insufficient evidence to support the proposed solution's efficacy in real-world arbitrage trading. Clear articulation of the problem statement, detailed explanation of the proposed solution, rigorous evaluation methodology, and empirical validation are essential to establish the credibility and significance of the research. There are some points that need further clarification and improvement:
1) The manuscript lacks clarity and precision in defining the problem statement and describing the proposed solution. The terms "long-term cointegration relationship," "significant price fluctuations," and "trend characteristics" need clear definitions and context to understand their implications in arbitrage trading.
2) While the manuscript highlights disruptions in cointegration relationships and their impact on arbitrage strategies, it fails to provide concrete examples or evidence of how these disruptions occur in real-world trading scenarios. Without clear elucidation of the problem, the significance of the proposed solution is diminished.
3) The manuscript introduces the Dynamic-LSTM Arb (DLA) strategy but lacks details on its architecture, functionality, and effectiveness in addressing the identified challenges. It does not explain how LSTM is employed to classify trend movements or how trading suspension decisions are made based on emerging trends.

Experimental design

The manuscript claims superiority of the DLA strategy over the benchmark strategy and the returns of the CSI 300 Index but does not provide empirical evidence or comparative analysis to support this assertion. Without robust benchmarking and validation, the claims made by the authors remain unsubstantiated.

Validity of the findings

1) While the manuscript mentions training results and theoretical returns achieved by the DLA model, it lacks specificity in terms of evaluation metrics and methodology. The absence of comparative analysis or statistical evidence undermines the credibility of the reported results.
2) “Discussion” section should be added in a more highlighting, argumentative way. The author should analysis the reason why the tested results is achieved.

Additional comments

The manuscript entitled “An LSTM-based optimization algorithm for enhancing quantitative arbitrage trading” has been investigated in detail. The manuscript suffers from vague terminology, lack of specificity, and insufficient evidence to support the proposed solution's efficacy in real-world arbitrage trading. Clear articulation of the problem statement, detailed explanation of the proposed solution, rigorous evaluation methodology, and empirical validation are essential to establish the credibility and significance of the research. There are some points that need further clarification and improvement:
1) The manuscript lacks clarity and precision in defining the problem statement and describing the proposed solution. The terms "long-term cointegration relationship," "significant price fluctuations," and "trend characteristics" need clear definitions and context to understand their implications in arbitrage trading.
2) While the manuscript highlights disruptions in cointegration relationships and their impact on arbitrage strategies, it fails to provide concrete examples or evidence of how these disruptions occur in real-world trading scenarios. Without clear elucidation of the problem, the significance of the proposed solution is diminished.
3) The manuscript introduces the Dynamic-LSTM Arb (DLA) strategy but lacks details on its architecture, functionality, and effectiveness in addressing the identified challenges. It does not explain how LSTM is employed to classify trend movements or how trading suspension decisions are made based on emerging trends.
4) While the manuscript mentions training results and theoretical returns achieved by the DLA model, it lacks specificity in terms of evaluation metrics and methodology. The absence of comparative analysis or statistical evidence undermines the credibility of the reported results.
5) The manuscript claims superiority of the DLA strategy over the benchmark strategy and the returns of the CSI 300 Index but does not provide empirical evidence or comparative analysis to support this assertion. Without robust benchmarking and validation, the claims made by the authors remain unsubstantiated.
6) “Discussion” section should be added in a more highlighting, argumentative way. The author should analysis the reason why the tested results is achieved.
7) The authors should clearly emphasize the contribution of the study. Please note that the up-to-date of references will contribute to the up-to-date of your manuscript. The studies named- “Crude oil time series prediction model based on LSTM network with chaotic Henry gas solubility optimization; Performance of metaheuristic optimization algorithms based on swarm intelligence in attitude and altitude control of unmanned aerial vehicle for path following; A new hybrid model for wind speed forecasting combining long short-term memory neural network, decomposition methods and grey wolf optimizer”- can be used to explain the optimization process and methodology in the study or to indicate the contribution in the “Introduction” section.
8) It will be helpful to the readers if some discussions about insight of the main results are added as Remarks.
This study may be proposed for publication if it is addressed in the specified problems.

Cite this review as

---

## Round 0.2 · Minor Revisions

As per comments from original reviewers, I suggest a minor revision for this revised paper.

Reviewer 2 ·

Basic reporting

The authors have diligently revised and improved the manuscript based on the feedback provided by reviewers during the previous review cycle. As a result of these revisions, the quality of the manuscript now aligns with the standards set by the journal. However, upon thorough examination of the manuscript, it becomes evident that there are certain linguistic and grammatical issues present throughout. To address these concerns, the authors are strongly advised to have the manuscript proofread by a fluent English speaker. This step will help rectify any lingering linguistic or grammatical errors and ensure that
the manuscript reads fluently and coherently. Additionally, it is imperative for the authors no meticulously review and confirm that all references cited in the manuscript adhere to the journal's prescribed referencing style and format. Ensuring consistency and accuracy in reference formatting is crucial for maintaining the integrity and professionalism of the manuscript. By taking these measures to address language and referencing issues, the authors can further enhance the clarity, readability, and overall quality of their manuscript, thereby facilitating a smoother publication process and better dissemination
of their research findings.

Experimental design

.

Validity of the findings

.

Additional comments

.

Cite this review as

Reviewer 3 ·

Basic reporting

My comments have been addressed. It is acceptable in the present form.

Experimental design

My comments have been addressed. It is acceptable in the present form.

Validity of the findings

My comments have been addressed. It is acceptable in the present form.

Cite this review as

---

## Round 0.3 · accepted · Accept

In regard to this revised paper, it can be accepted now.